# Glucose Levels as a Key Indicator of Neonatal Viability in Small Animals: Insights from Dystocia Cases

**DOI:** 10.3390/ani15070956

**Published:** 2025-03-27

**Authors:** Raquel Rodríguez-Trujillo, Miguel Batista-Arteaga, Kseniia Iusupova

**Affiliations:** Unit of Reproduction, University Institute of Biomedical Research and Health, University of Las Palmas de Gran Canaria, Transmontaña s/n, 35413 Arucas, Las Palmas, Spain; raquel.rodriguez@ulpgc.es (R.R.-T.); ksfedorchak23@gmail.com (K.I.)

**Keywords:** glucose, neonate, Apgar score, viability, mother

## Abstract

Neonatal mortality in small animals, such as dogs and cats, can be alarmingly high, especially in situations of difficulty during delivery. This study investigates how glucose levels in newborns can serve as a key indicator of their viability. We analyzed 54 mothers and their 284 neonates to understand the relationship between maternal weight, litter size, and neonatal glucose levels. Our findings indicate that a glucose level below 79.50 mg/dL is associated with a decreased likelihood of survival in newborns. Additionally, neonates with low birth weight and those from smaller litters showed a higher prevalence of hypoglycemia. The results highlight the importance of monitoring glucose levels in neonates, as hypoglycemia can lead to serious health issues. This research emphasizes the need for timely interventions to improve neonatal outcomes and reduce mortality in small animal litters, thereby benefiting veterinarians and pet owners alike.

## 1. Introduction

Neonatal mortality during the first years of life in small animals presents an alarming rate, reaching figures as high as 40%, which raises significant concern among both breeders and veterinarians [1]. Perinatal mortality, which ranges from 20% to 40%, is particularly high compared to other domestic species [2]. This phenomenon is attributed to the abrupt transition from the intrauterine to the extrauterine environment, necessitating rapid and effective adaptation by the neonates [3]. This adaptation is intrinsically linked to the development, maturity, and viability of the individuals at the time of birth [2].

The primary cause of neonatal mortality is associated with immaturity at the time of birth, rendering these individuals particularly vulnerable [1]. Mortality can manifest during formation in utero, during expulsion, immediately after birth, or in the first weeks of life, with the first week being the most critical [3]. At the time of birth, the most common causes of mortality include asphyxia, acidosis, glucose imbalance, and hypothermia [2,4,5]. However, during the neonatal period, infections, particularly those of bacterial origin, represent the second most common cause of mortality, following hypoxia [6,7]. Among the most characteristic clinical signs of neonates affected by infections is the neonatal triad, which includes hypothermia, hypoglycemia, and dehydration [6]. The depression and lethargy observed in these individuals can result in the loss of the suckling reflex, which in turn leads to decreased intake of colostrum or milk and, consequently, weight loss [6]. This situation may trigger hypoglycemia, which, according to some authors, occurs in 65.5% of neonates with bacterial infections [6]. Additionally, neonatal hypoxia and hypercapnia are consequences of pulmonary immaturity. However, these conditions can also develop due to umbilical cord occlusion during whelping. If this condition persists, it may lead to respiratory acidosis, accumulation of acid radicals, and subsequently to metabolic acidosis, decreased glucose levels, and bradycardia [8,9]

Neonatal viability refers to the condition and reactivity of the neonate, factors that influence its survival. A neonate is considered viable when it exhibits efficient breathing after birth, emits clear cries, actively seeks the mammary gland, is capable of properly grasping the nipple, and demonstrates normal neonatal reflexes, including the suckling reflex [10]. The Apgar score has been recognized as an effective method for assessing neonatal viability in various species, including humans, horses, cows [11], and dogs. This method allows for the classification of neonatal viability, and based on the obtained score, the prognosis for survival in the first 24–48 h can be determined [12]. Several authors have described classification ranges: a score of 7–10 points indicates a survival rate of 95–100% [10,13,14]; a score of 4–6 points suggests a survival rate of 94–100% [9]; and a score of 0–3 points is indicative of a survival rate of 0–39% [3,10,15].

Newborns have a predisposition to suffer from hypoglycemia, attributed to hepatic immaturity, limited energy reserves, and immaturity in glucose balance [15,16,17]. This situation, along with the low capacity for gluconeogenesis and glycogenolysis, impairs the ability to maintain glucose levels within the normal range during the first days of life [4]. Additionally, the loss of glucose through urine, low muscle mass, absence of adipose tissue, and limited capacity to utilize free fatty acids as an alternative energy source are factors that increase the risk of hypoglycemia [18].

Hypoglycemia may occur as a consequence of various neonatal pathologies, such as neonatal sepsis or endotoxemia, hepatic dysfunction, low birth weight, as well as maternal factors like placental abnormalities or gestational diabetes [10,19]. This disorder has been identified as one of the leading causes of neonatal mortality and presents characteristic symptoms that include neurological signs such as lethargy, absence of suckling reflex, depression, seizures, coma, and even death [10]. Initially, a cutoff value of 40 mg/dL was established as indicative of neonatal hypoglycemia [19]; however, subsequent studies have suggested different cutoff values. A value of 40 mg/dL was associated with a low Apgar score and decreased reflexes, whereas a value of 28 mg/dL was linked to a higher probability of mortality during the first 7 days of life [20].

Other authors have indicated that values below 37 mg/dL within the first 8 h of life are associated with an increased mortality risk in the first 24 h [21]. Furthermore, additional investigations have evaluated the influence of hyperglycemia in neonates, noting that elevated glucose levels correlate with an increase in neonatal mortality; additionally, lower Apgar scores are associated with a higher risk of mortality alongside higher blood glucose levels [1,22]. Nonetheless, the conclusion of several studies is that glucose measurement does not serve as a useful tool for predicting neonatal mortality [22,23,24]. During the first weeks of life, different ranges of glucose values can be observed, with values between days 1 and 3 ranging from 76 to 155 mg/dL, between days 8 and 10 from 101 to 161 mg/dL, and between the fourth and fifth week from 121 to 158 mg/dL [10]. Recent studies measuring glucose immediately after birth and before the intake of colostrum report a normal value of 73.16 mg/dL. However, this value may vary depending on the type of delivery, being generally higher in neonates born via cesarean section and in cases of dystocia, reaching levels above 120 mg/dL [10].

Neonatal hypoglycemia may also be associated with various neonatal conditions, such as low birth weight, which typically leads to an increased metabolism and consequently a greater energy requirement. In neonates with low birth weight, the liver is often smaller, implying that hepatic reserves are reduced, thereby predisposing them to hypoglycemia [25]. Additionally, factors such as breed may influence this predisposition, as smaller or “toy” breeds exhibit greater susceptibility compared to larger breeds [25].

The process of parturition generates a state of stress for both the mother and the fetus, which can result in an increase in glucose production due to processes such as glycogenolysis and gluconeogenesis [24]. Glucose levels in neonates subjected to stressful situations correlate directly with maternal glucose levels [24].

Further investigations have explored the concentrations of glucose in amniotic fluid, observing a correlation between these concentrations and an increased probability of mortality. A positive correlation has been indicated between low glucose concentration and a higher likelihood of mortality at the time of birth [12]. However, no correlation was found with other parameters, such as the Apgar score [12]. For this reason, several authors have described that glucose concentrations in amniotic fluid correlate more closely with the metabolic maturity of the neonate [12,23]. Moreover, immaturity at the time of birth has also been investigated in relation to the renal function of the neonate, through urinalysis [5]. These authors suggest that the presence of glucose in the urine may indicate alterations in the functioning of the renal tubules as well as in the reabsorption processes [5].

The objective of the present study was to analyze glucose levels and how they influence neonatal viability. Additionally, this study evaluated how maternal glucose and other factors could alter blood glucose concentrations. Currently, some authors have established cutoff values for neonatal glucose, showing differences among them; however, to the best of the authors’ knowledge, no study has measured glucose levels immediately after birth without neonatal food intake and under conditions of dystocia or fetal stress. Immediate glucose measurement is crucial in these neonates, as dystocia and fetal distress induce metabolic responses that can compromise viability. In these cases, hypoxia and activation of the sympathetic system may alter glucose homeostasis, affecting the neonate’s ability to adapt to extrauterine life. Evaluating glucose at this critical moment allows the identification of neonates at risk, optimizing clinical decision-making to improve survival rates. Furthermore, there are no authors who have assessed whether there are differences in the cutoff values between brachycephalic and non-brachycephalic neonates. Regarding the evaluation of maternal glucose concentrations, few authors have investigated how these levels change during gestation and how other factors, such as litter size, may influence them.

## 2. Materials and Methods

### 2.1. Animals

For this study, a total of 54 mothers along with their respective litters were included, representing a total of 284 neonates. The mothers were categorized according to their breed, age, and weight, and the size of the litter was considered after the procedure was performed.

Before carrying out the surgical procedure, the mothers underwent a physical examination in which parameters such as heart rate, respiratory rate, rectal temperature, and mucosal color were evaluated. Additionally, a complete blood analysis was performed, which included the measurement of glucose concentration.

The procedure was conducted at the Veterinary Clinical Hospital of the University of Las Palmas de Gran Canaria, over a two-year period of sample collection, from 2022 to 2024. The entire protocol was supervised and approved by the corresponding ethics committee, and the owners of the bitches were informed about the procedure prior to its execution.

All bitches included in this study were classified as ASA II (American Society of Anesthesiologists Physical Status Classification System), as they were not considered completely hemodynamically stable due to gestation and the modifications this entails, although they presented no functional limitations at the time of the intervention.

### 2.2. Experimental Design

The bitches were classified based on their weight and the size of the litter. For weight, two groups were defined: bitches weighing less than 10 kg *(n* = 22) and bitches weighing more than 10 kg (*n* = 32). Regarding litter size, three groups were established: litters of 1–2 neonates (*n* = 12), litters of 3–5 neonates (*n* = 27), and litters of more than 5 neonates (*n* = 15). Additionally, from the 54 participating mothers (*n* = 54), a total of 15 mothers were randomly selected for pre-surgical and post-surgical glucose measurements, with the aim of determining whether significant differences existed in the values after surgery.

A rigorous procedure was followed in the data collection of the mothers and the neonates to ensure that this did not impose additional stress or interfere with neonatal viability.

The neonates were also classified according to the same criteria as the mothers to assess whether there was maternal influence on the obtained results. Two groups were established based on maternal weight: neonates from mothers weighing more than 10 kg (*n* = 109) and neonates from mothers weighing less than 10 kg (*n* = 74). Likewise, depending on the size of the litter, three groups were formed: neonates from litters of 1–2 (*n* = 24), neonates from litters of 3–5 (*n* = 102), and neonates from litters of more than 5 (*n* = 158).

For all mothers, the same procedure was followed upon arrival at the clinic. First, after evaluating the physical examination, an ultrasonographic examination (C4-1 Curved Array, ZONE Sonography^®^, Mindray Zonare Z. One PRO, Mountain View, CA, USA) was performed to determine the presence of fetal stress. Subsequently, a blood sample (3 mL) was taken to conduct a complete blood count (Procyte dx, IDEXX Laboratories S.L, Barcelona, Spain), a biochemical analysis (Catalyst Dx, IDEXX Laboratories S.L, Barcelona, Spain), and to measure progesterone levels (Speed Reader, Speed™ Progesterone, VIRBAC Spain, S.A, Barcelona, Spain). Only those mothers with progesterone levels below 1 ng/dL, who exhibited evident signs of parturition (at least 30 min of uterine contractions without fetal expulsion, suggesting potential dystocia), or who showed signs of fetal stress (fetal heart rate below 180–160 bpm on ultrasound) at the time of examination, were selected for this study.

### 2.3. Pre-Anesthetic and Pre-Surgical Evaluation

Initially, a simple physical examination was conducted that was designed to avoid generating stress in the pregnant mother. During this examination, heart rate was measured using a stethoscope, respiratory rate was assessed, mucosal color was observed, and a thoracic radiograph (R108, Ralco S.R.L., Biassono, Italy) and an electrocardiogram (MAC600, GE Medical System information technologies Inc., Karnataka, India) were performed. Subsequently, an ultrasonographic examination was conducted to measure the heart rate of at least 50% of the fetuses, along with the collection of blood samples to determine the suitability of the animal for the surgical procedure.

While awaiting the results of the analyses, pre-oxygenation of the bitch was initiated for at least 10 min, and preparations for the surgical procedure commenced. A catheter was placed in the cephalic vein of the forelimb, and the operating room was prepared for surgery. Once it was confirmed that the bitch could safely and stably enter the operating room, intravenous premedication was administered with fentanyl (Fentadon 50 μg/mL, Northwich, UK).

In addition to preparing the surgical area, a recovery zone was established, where special attention was given to temperature control using electric blankets (Carbon Vet, B.Braun VetCare, S.A, Barcelona, Spain), thermometers (Digital Braun PRT1000, B.Braun VetCare S.A, Barcelona, Spain), light lamps, and towels. A stethoscope was also available to monitor heart rate, along with oxygen therapy and the necessary drugs for resuscitation.

### 2.4. Patient Preparation and Monitoring

Initially, the bitch was induced with propofol at a dose of 4 mg/kg (Propovet, 10 mg/mL, Esteve, Barcelona, Spain). Although the total dose was calculated based on the weight of the bitch, an initial dose of 1 mg/kg was administered, with a one-minute wait between each dose. Following induction, the bitch was intubated and connected to the respiratory machine with automatic ventilation (Ventilator GE Datex Ohmeda Anesthesia Machine, GE Medical System information technologies Inc., Karnataka, India). To maintain the anesthetic plane, sevoflurane at 2% (SevoFlo 250 mL, Zoetis Inc., Tokyo, Japan) was used, which was gradually reduced during the surgery.

Throughout the surgical procedure, automatic monitoring was conducted, which included heart rate, respiratory rate, blood pressure (systolic and diastolic), temperature, oxygen saturation, and carbon dioxide saturation (WATO EX-35Vet, Shenzhen Mindray Animal Medical Technology Co., Ltd., Shenzhen, China). Blood pressure was measured indirectly using a cuff on the limb, selected according to the size of its circumference. A pulse oximeter was placed on the patient’s tongue to measure oxygen saturation, while heart rate was monitored using electrodes attached to the pads of the bitch. Finally, temperature was measured with an oral thermometer (uMEC12 Vet, Shenzhen Mindray Animal Medical Technology Co., Ltd., Shenzhen, China), and carbon dioxide saturation, as well as respiratory rate, were evaluated using a capnometer (uMEC12 Vet, Shenzhen Mindray Animal Medical Technology Co., Ltd., Shenzhen, China). All measurements were updated every 5 min throughout the entire duration of the surgery, from start to finish.

During the surgery, fluids (Ringer Lactate^®^ 3 mL/kg/h, Braun, Barcelona, Spain) were administered to maintain blood pressure, along with a continuous rate infusion (CRI) of fentanyl (Fentadon 50 μg/mL, Northwich, UK) for pain control throughout the surgical procedure.

### 2.5. Surgical Procedure

For the cesarean sections, a standard procedure was performed, beginning with an incision in the medial abdominal area. The subcutaneous tissue was dissected to expose the linea alba, through which access to the abdominal cavity was gained. Subsequently, the uterine horns were exteriorized with care to avoid ruptures, as the uterus exhibits significant friability during gestation. An incision was then made in the body of the uterus, and the uterine horns were alternately massaged until all the neonates were extracted, which were then transported to the recovery area.

To conclude the surgery, the uterus was closed using a continuous simple suture with atraumatic absorbable monofilament suture material (Monosyn^®^ 3/0, HR22, B. Braun Surgical SA, Rubí, Spain), followed by the application of an inverted pattern (Cushing). An abdominal lavage was performed with a warmed saline solution, both within the abdominal cavity and the uterine horns. Finally, the abdominal cavity was closed in layers, and oxytocin (1–4 IU Oxiton^®^, Ovejero, León, Spain) was administered intravenously to facilitate the expulsion of uterine remnants and promote the descent of milk.

### 2.6. Neonatal Resuscitation

In the recovery area, the same veterinarians were consistently responsible for the resuscitation process and the collection of results. The same resuscitation protocol was utilized: the ABC protocol (airway/breathing/cardiac). Before commencing, the temperature was monitored using electric blankets (Carbon Vet, B.Braun VetCare, S.A, Barcelona, Spain) and light lamps, and the neonates were dried with towels.

To facilitate ventilation, suction of the contents from the mouths and noses of the neonates was initiated using a bulb syringe or neonatal suction device (Beaba, Paris, France), with the aim of eliminating fetal fluids and meconium. It was verified that the airways were completely cleared, and during this process, the neonates were vigorously rubbed with a towel, which also aided in the expulsion of airway contents. During this process, the head was carefully supported and slightly tilted to promote the elimination of fluids, avoiding any swinging or potential impact on the brain, which could lead to increased intracranial pressure, subdural hemorrhages, or aspiration of stomach contents into the airways (aspiration pneumonia).

Once the neonates exhibited continuous breathing, they were maintained on oxygen therapy during the Apgar evaluation and the determination of other measures (temperature, glucose, lactate, and weight). Additionally, throughout this time, the temperature was cautiously monitored (Digital Braun PRT1000, B.Braun VetCare S.A, Barcelona, Spain) due to the thermal loss that occurs during the first hour of life.

For those neonates who were not breathing adequately or who presented bradycardia (low heart rate), an additional resuscitation protocol was instituted, combining the use of medications and cardiac compressions. Two to three compressions were performed per second, and a catheter was placed in the jugular vein. If necessary, medications were administered in the following order: naloxone (0.05 mg/kg, Naloxona B. Braun 0.4 mg/mL, B. Braun Medical, SA, Barcelona, Spain) and epinephrine (0.2 mg/kg; Adrenalina B.Braun 1 mg/mL, B. Braun Medical, SA, Barcelona, Spain). The resuscitation process was maintained for 45 min, and if no favorable outcome was achieved, it was discontinued. Neonates with congenital malformations incompatible with life were euthanized.

### 2.7. Apgar Assessment and Data Collection

To determine neonatal viability, the Apgar test was performed. This procedure involves measuring several parameters: heart rate, respiratory rate, mucosal color, mobility, and reflexes/irritability. Heart rate was measured using a stethoscope, while respiratory rate was counted by assessing the number of spontaneous breaths per minute. Mucosal color was evaluated through direct visual examination; mobility was assessed based on spontaneous movements; and reflexes of irritability were determined by the neonate’s response to external stimuli, such as compression on the pads or skin. For each parameter, a score of 0, 1, or 2 points was assigned, resulting in a final score ranging from 0 to 10. Based on this score, neonatal viability was classified into three categories: critical neonates (<3 points), moderate viability (4–7 points), and normal viability (8–10 points). This assessment was conducted immediately after neonatal resuscitation, within the first 5–10 min of life. The order in which neonates were removed from the uterus was recorded, although it was not considered as a variable in this study.

Subsequently, each neonate was weighed individually using a scale, and temperature was measured with a thermometer. To determine glucose and lactate levels, a blood sample was obtained from the pad of the hind limb, which was analyzed using a glucometer (Advocate PetTest, Rafael del Campo, Córdoba, Spain) and a lactate meter (Cera Check Lactato, RAL S.A, Barcelona, Spain).

Once the neonates were stable and measurements had been completed, they were transferred to the incubator (Vetario© S40, Vetario, Weston-super-Mare, UK), where a temperature of 33–34 °C was maintained, and they were immediately fed, provided that their body temperature exceeded 34.5 °C. One of the strengths of this study is the strict control over neonatal feeding prior to glucose measurement. By keeping neonates separate from the mother and standardizing the first feeding through orogastric tube administration with a milk replacer, we minimized potential confounding factors related to maternal colostrum intake. This methodological approach ensures that the glucose values reported truly reflect the neonates’ metabolic state immediately after birth, rather than postprandial variations.

### 2.8. Statistical Analysis

Statistical analyses were conducted using SPSS Statistics and Excel 2019 (SPSS Inc., Chicago, IL, USA). Initially, the normality of the data was assessed using the Shapiro–Wilk test, which yielded statistically significant results, indicating that the data did not follow a normal distribution. Receiver Operating Characteristic (ROC) curve analysis was performed to determine the cutoff values, using the Youden Index to identify the optimal threshold. Sensitivity was calculated as TP/(TP + FN), where TP (true positive) represents the number of correctly identified positive cases, and FN (false negative) represents the number of positive cases incorrectly classified as negative. Similarly, 1-specificity (false positive rate) was obtained as 1 minus the specificity value, where specificity was calculated as TN/(TN + FP), with TN (true negative) representing the correctly identified negative cases and FP (false positive) representing the negative cases incorrectly classified as positive. All values were derived from the ROC curve analysis. Given the non-normal distribution of the data, Spearman’s rank correlation was used for correlation analyses. Variables with a *p*-value ≤ 0.05 were considered statistically significant.

To ensure that our sample size was adequate for detecting meaningful differences in glucose levels across groups, a power analysis was performed using a two-tailed t-test with equal variance assumed. The analysis indicated that a sample size of 26 neonates per group would be required to achieve a power of 0.8 with an effect size of 0.8 and a significance level of 0.05. Given that our study included 284 neonates, the sample size was sufficient to detect statistically significant differences in glucose levels.

## 3. Results

### Mother and Neonate Glucose

This study involved 54 mothers and 284 neonates, classified by maternal weight into two groups: under 10 kg (*n* = 22) and over 10 kg (*n* = 32). Mothers were also categorized by litter size: 1–2 neonates (*n* = 12), 3–5 neonates (*n* = 27), and more than 5 neonates (*n* = 15). Similarly, neonates were classified based on maternal weight (74 from mothers < 10 kg and 109 from mothers > 10 kg) and litter size (24 neonates from 1–2, 102 from 3–5, and 158 from more than 5). Neonatal mortality was recorded at 6% (*n* = 17), with 2.5% (*n* = 7) of live-born neonates dying within the first hours and 3.5% (*n* = 10) being stillborn. Maternal mortality was 0%.

Table 1 compares the mean blood glucose levels among the mothers, along with the standard deviation for those weighing more than 10 kg and for those weighing less than 10 kg. Additionally, the mean glucose levels of the neonates were presented along with their corresponding standard deviation, based on the mothers’ weight. When comparing the glucose levels of the mothers according to their weight, no significant differences were observed between the groups. However, concerning the neonates, significant differences (*p* < 0.05) were found between both groups.

Additionally, a comparison was made between groups based on litter size regarding the mean blood glucose levels in the mothers (Table 2). The three groups were compared using the Kruskal–Wallis test for independent variables, and it was observed that there were no significant differences (*p* > 0.05) in the glucose levels of the mothers according to litter size. However, it was noted that the mean glucose level was slightly lower in the bitches with 3–5 neonates compared to the other two groups.

Regarding the glucose levels in the neonates, an analysis was conducted to determine if there were differences based on litter size (Table 2). The mean glucose level was slightly higher in bitches with 1–2 neonates compared to the other two groups. However, the differences observed between groups were not significant (*p* > 0.05), according to the Kruskal–Wallis test.

On the other hand, an investigation was conducted to determine whether there were differences in the glucose levels of the mothers before and after the surgical procedure. The results indicated that following surgery, the mean glucose level was slightly higher (120.93 ± 33.54) compared to the levels before surgery (114.79 ± 31.84). However, no significant differences were observed between the groups.

Of the total neonates, those that were brachycephalic (*n* = 84) were selected to establish a cutoff for glucose based on the obtained Apgar score. The ROC curve showed a good capacity of the model to distinguish between positive cases (Apgar > 7 points) and negative cases (Apgar < 7 points), as the curve was situated above the reference line (diagonal line). The curve exhibited a rapid ascent, indicating a high true positive rate (Figure 1). The ROC curve had an area under the curve (AUC) of 0.882, indicating good model performance. This value suggested that the model was capable of effectively distinguishing between positive and negative classes. The standard deviation was 0.060, and the significance level was <0.001, indicating that the results were statistically significant. The 95% confidence interval for the AUC ranged from 0.765 to 0.998, demonstrating that the model was consistently effective across different samples. Based on these data, a cutoff of 89.50 mg/dL was established (sensitivity 98.1%; specificity 81.5%).

Regarding Figure 2, a precision–recall curve was presented, indicating the model’s performance and its classification capability based on both parameters. Sensitivity measured the proportion of positive cases correctly identified by the model, while precision indicated the accuracy in predicting positives. The curve started at the origin and moved towards the upper right corner, reflecting good overall model performance. However, a sharp decline in precision was observed near the far right end, suggesting that, in attempting to maximize sensitivity, precision was slightly compromised.

On the other hand, the ROC curve (Figure 3) was represented for non-brachycephalic neonates (*n* = 111) concerning glucose levels based on neonatal Apgar scores. The obtained area under the curve (AUC) was 0.842, indicating good model performance in classifying positive cases (Apgar > 7 points) and negative cases (Apgar < 7 points). The curve rose rapidly towards the vertical axis, suggesting a high true positive rate and a low number of false positives. The standard deviation was 0.064, and the significance level was <0.001, indicating that the results were skmtatistically significant. The 95% confidence interval ranged from a minimum of 0.717 to a maximum of 0.968, reinforcing the reliability of the model. Based on these data, a cutoff of 87.50 mg/dl was established (sensitivity 97.5%; specificity 83.9%).

Additionally, the sensitivity–precision curve (Figure 4) was represented, starting near the origin and rising, indicating that the model improved its performance as sensitivity increased. The curve stabilized at a high level of precision, suggesting that the model maintained a good balance in correctly identifying positive cases and minimizing false positives. However, towards the end, a significant drop occurred, which may have implied that precision was compromised when attempting to maximize sensitivity.

In relation to Figure 5, the ROC curve was presented to determine the cutoff point for the total neonates in this study (*n* = 282). The model showed a good capacity to distinguish between classes, as the curve was positioned above the reference line (diagonal line). The curve exhibited a rapid initial growth, suggesting a high true positive rate and a low number of false positives. The area under the curve (AUC) was 0.710, indicating moderate model performance between the classes. The standard deviation was 0.045, and the significance level was <0.001, indicating that the results were statistically significant. The 95% confidence interval ranged from 0.622 to 0.799, implying that although the model performed better than a random one, there was room for improvement in its precision and discrimination. The obtained cutoff was 79.50 mg/dL (sensitivity 99.2%; specificity 59.2%).

On the other hand, a precision–recall curve (Figure 6) was constructed for the same model, showing that the curve started at the origin and rose rapidly, indicating that the model improved its precision with an initial increase in sensitivity. As it progressed, the curve stabilized at a precision of 0.8, suggesting that the model exhibited better precision in that region. However, as it approached higher sensitivity levels, there was a decline in precision, indicating that the model struggled to maintain prediction quality.

Neonatal glucose was correlated with other neonatal parameters, yielding the following results. When glucose was correlated with the Apgar score, a weak correlation was observed (*rs* = 0.3), which was statistically significant (*p* < 0.0001). However, when glucose was correlated with lactate and temperature, a very weak correlation was found (*rs* = 0.12 for both parameters), which did not reach statistical significance. Finally, regarding the neonate’s weight, no correlation was observed between the two parameters (*rs* = −0.004), which was also not statistically significant.

## 4. Discussion

Neonatal mortality has been the subject of study by various authors, showing differences among different studies, although it commonly hovers around 20% [21]. This mortality rate tends to increase in the first weeks of life. In our study, neonatal mortality was 6%, of which 3.5% corresponded to neonates that were stillborn and 2.5% to those that were born alive but died within the first hours of life. This discrepancy in percentages, with a higher proportion among stillborn neonates, may be related to the fact that all cesarean sections performed were urgent and not scheduled, and some presented dystocia upon arrival at the Veterinary Clinical Hospital. Conversely, other authors have reported lower mortality rates, around 11–13% [16] or even as low as 4% [26]. Neonatal mortality is a significant challenge for veterinarians and breeders. This study aimed to analyze the impact of neonatal markers like glucose, lactate, and birth weight on births during dystocia cesarean sections or fetal stress. It also established cutoff points based on the Apgar score and examined the influence of glucose on other neonatal and maternal factors.

Neonates are particularly susceptible to hypoglycemia due to limitations in their glycogen energy reserves and their minimal capacity for gluconeogenesis [6,8]. Their ability to maintain normoglycemia is limited, and hepatic immaturity in neonates is not sufficiently efficient to generate energy [27,28]. For this reason, some authors have indicated that low glucose levels (<92 mg/dL) are associated with a higher risk of mortality within the first 24 h of life [6]. In our study, the mean glucose level of the neonates (*n* = 282) was 115.6 mg/dL. This result could explain our low neonatal mortality both in the first hours and during the first week of life. In our study, glucose was measured immediately after birth, as neonatal glucose levels tend to drop rapidly within the first 24 h of life [22]. This immediate measurement is particularly relevant in neonates born via cesarean section due to dystocia, as the metabolic transition in these individuals may be altered by factors such as fetal hypoxia, oxidative stress, and an exacerbated endocrine response. Studies in human neonates have shown that perinatal stress can induce transient hyperglycemia or, in more severe cases, persistent hypoglycemia, compromising neonatal viability [29,30]. Our findings reinforce the importance of assessing glucose at this critical moment to detect metabolic alterations early and improve clinical intervention strategies. However, other authors [21] have reported a mean value of 97 mg/dL in measurements taken between the first 10 min and 8 h of life of the neonates. The author notes that these results are inconclusive, as the measurements were made on the total number of neonates without differentiating those who had taken milk. This slight increase in the mean glucose in our study may be related to the type of delivery, in this case, emergency or dystocia cesarean sections. Such circumstances may lead to increased hypoxia, which in turn elevates catecholamines, promoting the production of epinephrine and norepinephrine, which suppress insulin production and stimulate hepatic glucose release [24,31]. Therefore, it is plausible that there is an increase in the mean glucose level in our study compared to other authors.

Despite the mean glucose level in our study being 115.6 mg/dL, the results obtained when establishing the cutoff points according to the Apgar score showed significant differences. The Apgar score has been the subject of study by several authors, who have associated the obtained score with neonatal viability, suggesting that a higher score correlates with a greater probability of survival [32]. Some studies indicate that a score below 5 points compromises neonatal viability [13], while others establish a cutoff at 6 points as the optimal score to predict neonatal survival [21]. In this context, and considering a cutoff of 6 points, the present study determined the glucose cutoff for brachycephalic neonates, non-brachycephalic neonates, and the total neonates.

In the case of brachycephalic neonates (*n* = 84), it was observed that the cutoff for achieving an Apgar score greater than 7 points, and consequently better neonatal viability, is set at 89.5 mg/dL. On the other hand, in non-brachycephalic neonates (*n* = 111), the cutoff was established at 87.50 mg/dL. Although numerous studies address the mean glucose levels at birth [10,21,24], none have determined a cutoff based on neonatal viability as represented by the Apgar score. However, the overall result (both brachycephalic and non-brachycephalic neonates; *n* = 282) was 79.50 mg/dL.

The observed differences between groups could be explained by several theories. Firstly, the discrepancy in results between brachycephalic and non-brachycephalic neonates may be related to the level of hypoxia at birth. Lower Apgar scores are often correlated with reduced oxygen saturation, which tends to resolve as spontaneous breathing stabilizes [3]. Brachycephalic neonates face greater difficulties in achieving optimal scores due to their physiological conditions [33], which could justify a slightly higher glucose cutoff; by using the same scale for all, brachycephalic neonates may differ in the scores obtained. Additionally, other authors have noted specific modifications in the scale (such as mucosal color and heart rate), adapting it to the characteristics of brachycephalic neonates [13,32].

On the other hand, the authors of [20] have indicated that neonates with glucose levels below 40 mg/dL typically present lower Apgar scores and decreased reflexes [20]. However, according to the results obtained in our study, we consider that a glucose value lower than 79.50 mg/dL is associated with a significantly lower Apgar score compared to the rest of the neonates, which impairs their ability to suckle or respond to external stimuli, thus compromising their survival. Furthermore, other authors [3], reported that their mean glucose value was even lower, observing that neonates with a score below 7 had glucose levels of less than 30.5 mg/dL. Additionally, some studies indicate that a glucose concentration below 40 mg/dL begins to manifest visible clinical symptoms in the neonate [20]. A possible difference between the values reported by other authors and those obtained in our study may be related to the sample size. Another theory to consider is fetal stress; studies in humans have shown that neonates experiencing fetal stress at the time of birth present elevated glucose concentrations [3,34]. This may explain why our values are higher, given that the surgeries performed were urgent or dystocia-related.

Additionally, no significant correlation was observed between glucose concentration and neonatal mortality, as there were neonates who died with glucose levels below 50 mg/dL and others who died with concentrations exceeding 230 mg/dL. Therefore, as noted by other authors [35], no correlation is established between these two factors. However, glucose can be an important indicator of neonatal viability, as a neonate with hypoglycemia may indicate underlying pathologies such as sepsis or portosystemic shunts [10,15]. Nonetheless, the absence of correlation in our study may be attributed to the low mortality observed throughout this study, as described by Greghi et al. [35].

During gestation, fetuses maintain their glucose levels through continuous infusion via the placenta, without relying on gluconeogenesis [36]. A correlation has been described between maternal glucose and the glucose in the amniotic fluid of fetuses; furthermore, the glucose in the amniotic fluid has also been correlated with the blood glucose of the neonate obtained through the umbilical cord [32]. However, no direct correlation has been observed between maternal glucose and neonatal glucose. Therefore, one of the objectives of this study was to determine how maternal parameters (maternal glucose and maternal weight) influence neonatal glucose.

In our study, maternal glucose levels were measured before surgery, and neonatal glucose was assessed immediately after birth. We acknowledge that maternal glucose levels may fluctuate rapidly due to perioperative stress, anesthesia, and surgical interventions, potentially creating a temporal mismatch between maternal and neonatal glucose values. While our findings provide valuable insights into neonatal glucose dynamics, future studies should incorporate continuous maternal glucose monitoring before, during, and after surgery to better assess the relationship between maternal and neonatal glucose levels. Additionally, measuring neonatal glucose at multiple time points post-birth could help elucidate how glucose metabolism evolves in the immediate neonatal period. However, our results could be explained by several reasons. Firstly, other authors have indicated that neonatal glucose levels depend exclusively on the metabolic maturity of the neonate [12]. As mentioned earlier, glucose levels at birth depend on metabolic reserves and the neonate’s capacity for gluconeogenesis [6], which constitutes the primary reason for variation in glucose concentrations without direct influence from maternal glucose. Secondly, other neonatal factors, such as hypoxia, can influence glucose levels [31]. Hypoxia or asphyxia at the time of birth depends on the fetal stress experienced but is not directly related to maternal parameters. Therefore, this factor will influence the neonate’s glucose concentrations independently of the levels in the mother. A limitation of our study was not determining the oxygen saturation at the time of birth, which would have allowed us to observe if there was a correlation between both parameters.

Amniotic fluid analyses have begun to be a subject of study in veterinary medicine [12,32], similar to research in human medicine [15]. The conclusions of these studies indicate that there is a correlation between glucose levels and neonatal survival, as lower concentrations of glucose in the amniotic fluid are associated with reduced survival rates [15]. However, the glucose present in the amniotic fluid is primarily related to maternal metabolism and, to a lesser extent, to fetal metabolism, which is more influenced during advanced stages of development [15]. This could explain why there are studies that have demonstrated a correlation between maternal glucose and glucose in the amniotic fluid, but not with neonatal glucose levels at the time of birth.

Regarding maternal glucose at the time of delivery, some authors have described that glucose concentration is lower in small-breed bitches compared to large breeds [37]. In our case, no significant differences were found; however, the concentration was slightly higher in large breeds compared to small ones. Furthermore, glucose concentration at the time of delivery is related to the type of diet [37,38]. Other studies have observed correlations between neonatal glucose and maternal age [15] or have noted the possibility of variations in glucose concentration among breeds, similar to what occurs with other biochemical parameters, such as creatinine or ALT (alanine aminotransferase) [32].

In this study, we also attempted to evaluate whether litter size influenced glucose concentrations in the mother or neonate. No differences in concentrations were found among the three groups analyzed. To the best of the authors’ knowledge, there are no previous studies that have sought to identify differences in glucose concentration based on litter size. Only one study noted that mothers with large litters had a higher probability of hypoglycemia due to inadequate nutrient intake, which was attributed to the pressure exerted by the uterus on the stomach and its inability to distend properly [37,39].

The authors of [24] have determined the differences in cortisol and glucose levels in mothers based on the type of delivery, observing that levels were slightly higher in those undergoing cesarean sections. In our study, glucose concentrations were evaluated both before and after the surgery, and these values were slightly higher compared to those determined in mothers with natural births [24]. However, to the best of the author’s knowledge, no studies have evaluated maternal glucose concentrations before and after surgery. In our case, no significant differences were observed, although it was noted that the concentration was slightly higher after the surgical procedure. This could be attributed to the stress situation associated with cesarean sections, which leads to an increase in cortisol levels and, consequently, glucose levels [24,34].

Regarding other markers of neonatal viability (temperature, birth weight, Apgar score, or lactate), correlations with glucose at the time of birth were performed. In our case, the correlations were weak or not statistically significant. Other authors have sought correlations between glucose and birth weight, Apgar score, or lactate [1,3,6], but not with temperature. In relation to birth weight and lactate, consistent with our results, no correlation with glucose concentration was observed [1,6]. Regarding the Apgar score, we found a weak correlation; in contrast, the authors of [3] indicated a negative correlation between the two variables (higher glucose and lower Apgar score) [3]. However, other authors state that it is not a definitive indicator of neonatal morbidity or viability [24,25].

Currently, there are no studies correlating temperature with glucose, although all authors note that the occurrence of hypothermia and hypoglycemia is common. In our study, no relationship between these two factors was found at the time of birth [6,21,32]. This can be explained by the following reasons: hypothermia often causes neonatal depression, leading to the loss of the suckling reflex, decreased reflexes, and alterations in the cardiovascular system. Consequently, this prevents the neonate from ingesting milk, favoring the subsequent onset of hypoglycemia, which could lead to both conditions presenting simultaneously in critical neonates [6,10,33,40]. However, at the time of neonatal resuscitation, our study primarily monitored temperature, as neonates are unable to thermoregulate at birth. Therefore, this could be a limitation, as the temperature recorded during the measurement depended on the ambient temperature provided (e.g., heating pads, towels, light lamps, etc.).

## 5. Conclusions

This study highlights the critical role of glucose levels in neonatal viability, particularly in small animals undergoing cesarean sections due to dystocia. The findings reveal that neonatal hypoglycemia is a significant contributor to mortality within the first hours and days of life, underscoring the necessity for vigilant monitoring of glucose levels in neonates.

The established cutoff values for glucose, based on Apgar scores, demonstrate that lower glucose concentrations correlate with poorer neonatal outcomes, thereby emphasizing the importance of timely interventions in cases of hypoglycemia. Specifically, a glucose level below 79.50 mg/dL is associated with a marked decrease in viability, which can severely impair a neonate’s ability to suckle and respond to environmental stimuli.

Furthermore, this study indicates that maternal factors, such as weight and glucose levels, influence neonatal glucose status, though a direct correlation between maternal and neonatal glucose levels was not established. This suggests that while maternal health is pivotal, neonatal metabolic maturity and response to stressors also play a crucial role in determining glucose levels at birth.

Overall, these findings advocate for the implementation of routine glucose assessments in neonates immediately after birth, particularly in high-risk situations such as cesarean deliveries or when signs of fetal distress are present. By addressing hypoglycemia proactively, veterinarians can enhance neonatal survival rates and improve overall outcomes for small animal litters. Future research should continue to explore the intricate relationships between maternal and neonatal health parameters to further refine strategies aimed at reducing neonatal mortality in small animals.

## Figures and Tables

**Figure 1 animals-15-00956-f001:**
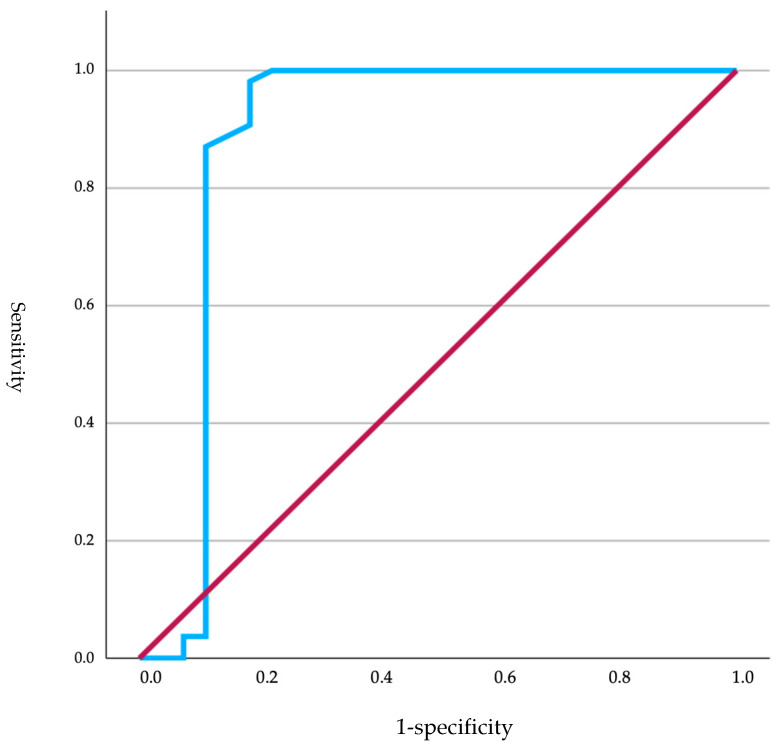
ROC curves analyzing glucose levels in relation to Apgar scores in brachycephalic neonates, where the red line represents the reference line (random performance) and the blue line represents the actual ROC curve reflecting the model’s performance.

**Figure 2 animals-15-00956-f002:**
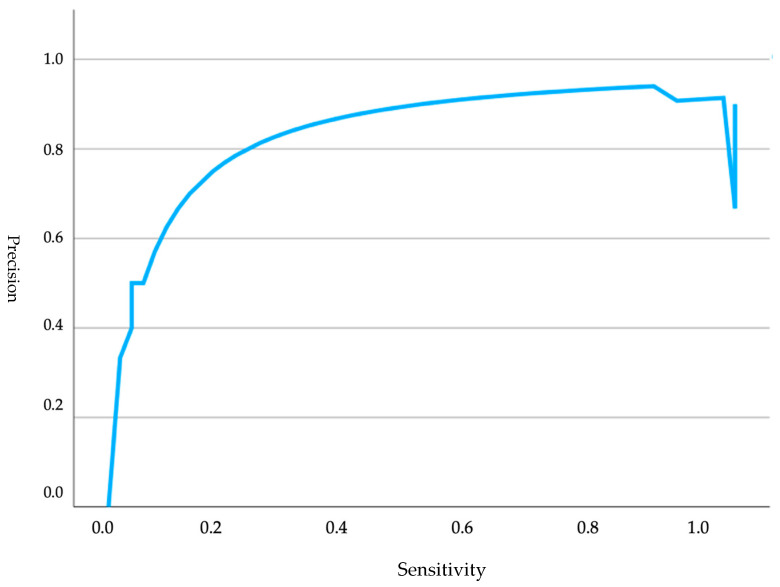
Precision curves analyzing glucose levels in relation to Apgar scores in brachycephalic neonates.

**Figure 3 animals-15-00956-f003:**
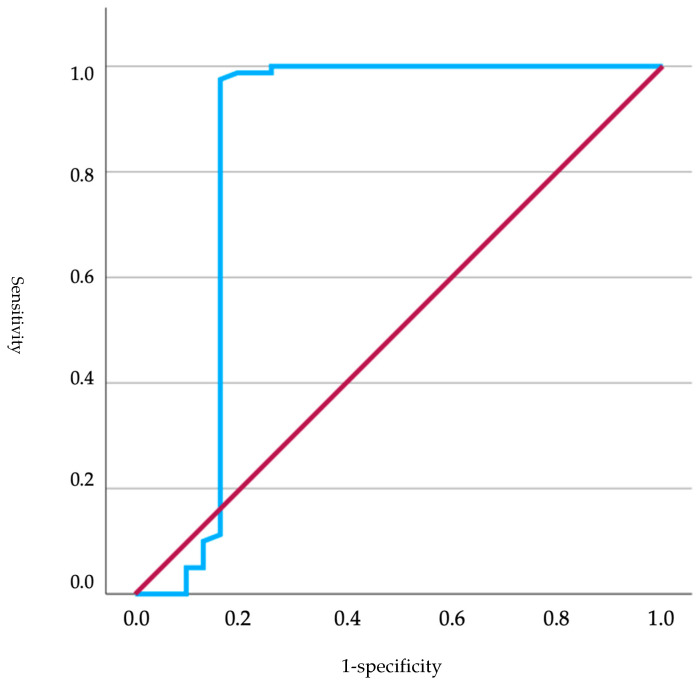
ROC curves analyzing glucose levels in relation to Apgar scores in non-brachycephalic neonates, where the red line represents the reference line (random performance) and the blue line represents the actual ROC curve reflecting the model’s performance.

**Figure 4 animals-15-00956-f004:**
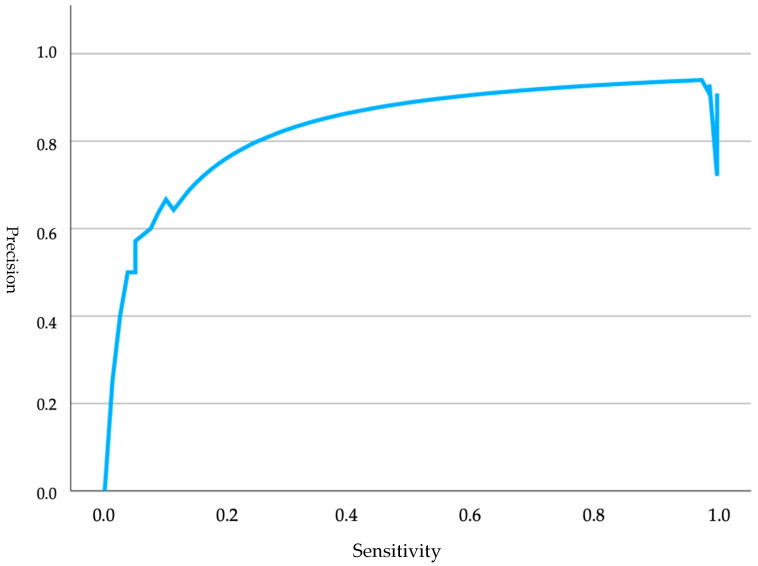
Precision curves analyzing glucose levels in relation to Apgar scores in non-brachycephalic neonates.

**Figure 5 animals-15-00956-f005:**
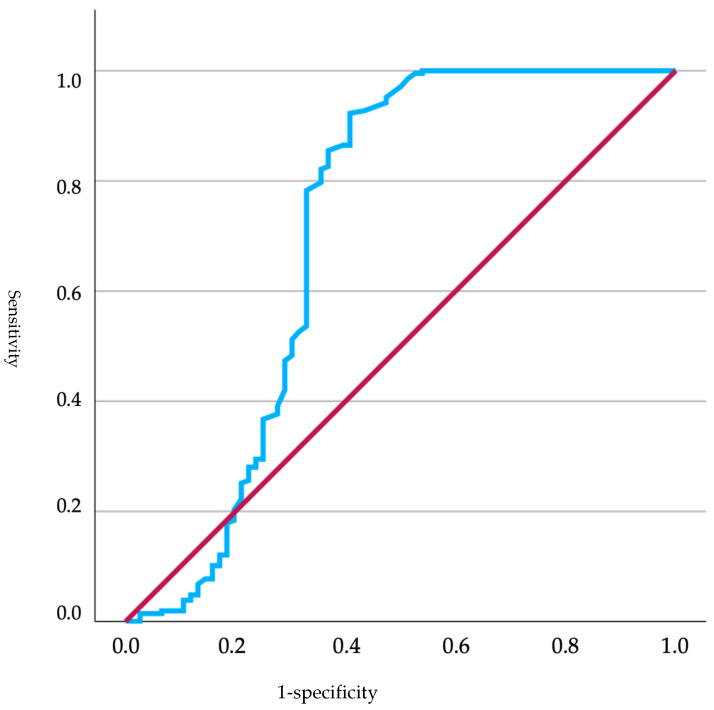
ROC curves analyzing glucose levels in relation to Apgar scores in neonates, where the red line represents the reference line (random performance) and the blue line represents the actual ROC curve reflecting the model’s performance.

**Figure 6 animals-15-00956-f006:**
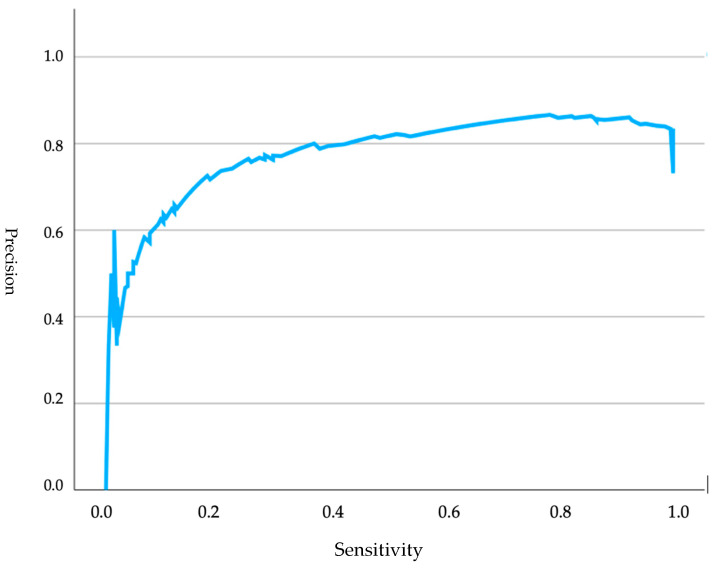
Precision curves analyzing glucose levels in relation to Apgar scores in neonates.

**Table 1 animals-15-00956-t001:** Mean (± SD) maternal and neonatal blood glucose concentrations depending on the weight.

	More Than 10 kg	Less Than 10 kg
Maternal glucose concentration	113.90 ± 20.80	108.79 ± 25.64
Neonatal glucose concentration	$110.59\;\pm \;7.69^{\;a}$	$130.40\;\pm \;15.37^{\;b}$

^ab^ Different letters in the same column denote significant differences in glucose concentration according to maternal weight (*p* < 0.05).

**Table 2 animals-15-00956-t002:** Mean (± SD) maternal and neonatal blood glucose concentrations according to litter size.

	Group	Mean	Std Err	Lower	Upper
Mother	1–2 neonates (*n* = 13)	114.73 ± 8.30	8.30	98.05	131.40
	3–5 neonates (*n* = 26)	110.04 ± 5.40	5.40	99.19	120.89
	>5 neonates (*n* = 15)	113.93 ± 7.35	7.35	99.15	128.71
Neonates	1–2 neonates (*n* = 13)	123.04 ± 10.36	10.36	102.66	143.43
	3–5 neonates (*n* = 26)	114.38 ± 5.02	5.024	104.49	124.27
	>5 neonates (*n* = 15)	114.98 ± 4.05	4.049	107.02	122.96

## Data Availability

Data will be available for all readers upon request.

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
