# Peer review of "Glucose Levels as a Key Indicator of Neonatal Viability in Small Animals: Insights from Dystocia Cases"

_animals, 2025, doi:10.3390/ani15070956_

Round 1
Reviewer 1 Report
Comments and Suggestions for Authors
The objective of the present study was to analyze glucose levels and their effect on neonatal viability. Glucose levels immediately after birth were also measured without the neonates having ingested food and under conditions of dystocia or fetal stress. The cutoff values between brachycephalic and non-brachycephalic neonates were also evaluated. Regarding the change during gestation in maternal glucose concentrations, it was also examined how other factors, such as litter size, may influence them. In this respect, the manuscript can be greatly supported. However, before doing it, the manuscript needs some clarification:
- Lines 174 and 176: “Only those mothers with progesterone levels below 1 ng/dL, who exhibited evident signs of parturition (at least 30 minutes of contractions), or who showed fetal stress at the time of the ultrasound, were selected for the study”.
- “exhibited evident signs of parturition (at least 30 minutes of contractions)” Does this mean that dogs exhibited at least 30 minutes of uterine contractions without any progressions?
- Please give the criteria for fetal stress.
- Lines 267 and 268: “This assessment was conducted immediately after the birth of the neonates”
- Does it mean that the APGAR scores of newborn puppies were evaluated immediately after birth or after resuscitation, as mentioned in the Materials and methods?
- Why was the APGAR score or glucose concentration not evaluated according to the order of the puppies removed from the uterus?
- Why was the effect of the length of gestation not evaluated in the study?
Line 53: “neonatal hypoxia and asphyxia are consequences of pulmonary immaturity”. Correctly: neonatal hypoxia and hypercapnia are consequences of pulmonary immaturity. However, neonatal hypoxia and hypercapnia can also develop because of occlusion of the umbilical cord during whelping.
Line 61: “viability in various species, including humans, horses, and dogs.” Correctly: viability in various species, including humans, horses, cows (there is much information about newborn calves- please give a review article), and dogs.
Line 84: [1616]. Correctly: [16]???
Line 165: ASA II (abbreviation, please give it)
Line 155: “(N=54)”. Correctly: (n=54)
Lines 155 and 156: “neonates from litters of 1-2 (n=24), neonates from litters of 3-5 (n=102), and neonates from litters of more than 5 (n=157)”. Totals: n=283. Line 289: The study involved 284 neonates. Lines 10 and 22: also n=284. Which one is correct?
Line 168: “an ultrasound”. Correctly: an ultrasound (or ultrasonographic) examination. Lines 176, 181: the same.
Lines 181 and 181: “a thoracic radiograph and an electrocardiogram were performed”. Please give the equipment types and manufacturers.
Line 191: “electric blankets, thermometers”. Please give the type and manufacturer.
Line 200: sevoflurane: the same.
Lines 206 and 207: pulse oximeter: the same.
Lines 209 and 210: oral thermometer, capnometer: the same.
Line 221 and 222: “An incision was then made in the cervix of the uterus.” Do you mean the body of the uterus?
Line 237: bulb syringe or neonatal suction device: the same
Lines 253 and 254: naloxone, epinephrine and heparin: the same
Line 278: Análisis estadístico: Please give it in English
Line 284: sensitivity and 1-specificity: Please describe how sensitivity and 1-specificity were evaluated. What is 1-specificity?
Line 290: under 10 kg (n=20) and over 10 kg (n=32). This is n=52: what happened with two mothers? Please give it.
Lines 292 and 293: neonates were classified based on maternal weight (74 from mothers <10 kg and 109 from mothers >10 kg). This is n=173. Later n=283 (Line 293). What was the reason for this remarkable difference?
Table 1:
Title: Correctly: Mean (±sd) maternal and neonatal blood glucose concentrations depending on the weight.
“Mother glucose, Neonate glucose” Correctly: Maternal glucose concentration, Neonatal glucose concentration
“ab Different letters in the same file and parameter (blood pressure/heart rate) denote significant differences (p < 0.05)”. What does blood pressure/heart rate mean here?
Line 318: Table 2: “This is a table. Tables should be placed in the main text near to the first time they are cited”. This is not the title for Table 2. Please give it.
Table 2: Mother = 51, Line 91: Mother = 54. Why are three mothers missing?
Lines 332 and 333: “the significance level was 0.000”: This is very strange. Line 258: the same.
Line 351: Correctly: Figures 1 and 2.
Line 372: Correctly: Figures 3 and 4.
Line 373: Correctly: in Non-Brachycephalic Neonates.
Line 394: Correctly: Figures 5 and 6.
Lines 479 and 484: Correctly: “by Greghi et al. [28]”.
Lines 500 and 501: Correctly: Secondly, other neonatal factors, such as hypoxia, can influence glucose levels [24].
Line 522: ALT (abbreviation, please give it)
Comments on the Quality of English Language
I am not an English expert but it can be improved.
Reviewer 2 Report
Comments and Suggestions for Authors
Dear Authors, thanks for eliciting me to review your work, these are my recommendations:
- The introduction could benefit from a more focused description of the existing literature on glucose as a biomarker for neonatal viability.
- A more precise justification for focusing on glucose measurements immediately after birth, particularly in dystocia cases, would strengthen the rationale.
- The word: Analisis estadístico is a mistake; please review all manuscript.
- The study does not sufficiently address the temporal mismatch between maternal glucose levels (pre-surgery) and neonatal glucose measurements (immediately after birth). Given that maternal glucose levels may change rapidly due to stress or surgery, comparing them without considering these factors could be misleading.
- There is no mention of a power analysis to determine if the sample size (54 mothers and 284 neonates) is adequate for detecting meaningful differences in glucose levels across groups. Given the variability of neonatal health outcomes, a power analysis would have strengthened the findings.
- Please clarify the timing of glucose measurements in neonates and address how milk intake was controlled.
- In the future, you could consider including a control group (non-dystocia cases) to further evaluate the role of glucose as a predictor of neonatal viability in various conditions.
The word: Analisis estadístico is a mistake; please review it in all manuscript.
